## PERSPECTIVE

# Don't forget the mucus barrier in pulmonary drug delivery!

Anita Balázs[1,2,3] (ID)
and Marcus A. Mall[1,2,3] (ID)

[1]*Department of Pediatric Respiratory Medicine, Immunology and Critical Care Medicine, Charité - Universitätsmedizin Berlin, Berlin, Germany*
[2]*German Center for Lung Research (DZL), associated partner site Berlin, Berlin, Germany*
[3]*German Center for Child and Adolescent Health (DZKJ), partner site Berlin, Berlin, Germany*

Email: marcus.mall@charite.de

Handling Editors: Peying Fong & Péter Hegyi

The peer review history is available in the Supporting Information section of this article (https://doi.org/10.1113/JP288762#support-information-section).

Primary human bronchial epithelial cell (HBE) cultures grown at an air–liquid interface recapitulate key components of the airway epithelium *in vivo* including transepithelial ion and fluid transport, secretion of the mucins (MUC5B and MUC5AC) that form the mucus layer, and coordinated beating of motile cilia. These functionalities are essential for proper mucociliary clearance, an innate defense mechanism that is crucial to clear constantly inhaled pathogens and pollutants from the lungs and fails in cystic fibrosis (CF) and other muco-obstructive lung diseases (Hill et al., 2022). Based on these properties, HBE cultures have emerged as the gold standard for studies of airway (patho)physiology, pharmacology and preclinical testing of novel therapeutic strategies. This is exemplified by CF, a severe genetic disease caused by mutations in the *CFTR* gene encoding a cAMP-dependent epithelial chloride channel, where studies in HBE cultures provided key insights into how CFTR dysfunction is linked to the development of highly viscoelastic mucus, impaired mucociliary clearance and chronic airway disease (Hill et al., 2022). Subsequently, studies in HBE cultures from patients with CF were critical for the successful development of small-molecule CFTR modulator drugs that restore the chloride channel function of the mutated CFTR protein and provide unprecedented benefits for currently up to 90% of CF patients who are eligible for these therapies (Graeber & Mall, 2023). Besides pharmacological activation, specific inhibitors are indispensable tools for defining ion channel function in health and disease. In this context, the potent and specific inhibitor CFTRinh-172 has been instrumental in quantifying CFTR-dependent ion transport and determining the response of mutant CFTR to pharmacological correction in electrophysiology studies.

In this issue, Guidone et al. (2025) found that, in contrast to highly effective inhibition of CFTR chloride channels in heterologous cells, cAMP-dependent chloride secretion across HBE cultures is only partially inhibited by CFTRinh-172, raising questions about its potency and the potential contribution of other ion channels in primary airway epithelial cells. The authors also observed that the effects of CFTR inhibitors are further diminished when HBE cultures are pre-treated with pro-inflammatory cytokines such as IL-17/TNF-$\alpha$ or IL-4 that increase mucin secretion. Based on these findings it was hypothesized that the mucus layer may hinder access for CFTR inhibitors to CFTR channels expressed at the apical cell membrane, and may thereby be responsible for their limited inhibitory effects. To test this hypothesis Guidone and colleagues used the reducing agent dithiothreitol (DTT) that lyses mucus by reducing disulfide bonds between mucins. Using this approach, the authors elegantly demonstrate that removal of the mucus layer on HBE cultures greatly increases the fraction of the cAMP-dependent chloride current that is sensitive to CFTR inhibitors (Guidone et al., 2025). Collectively, these results convincingly show that the mucus layer forms an important barrier for pharmacological targeting of the airway epithelium.

These data have important implications for CF and beyond, as they highlight the importance of the mucus layer covering airway surfaces not only as a barrier that protects the lungs from potentially harmful environmental stimuli, but also as an important barrier to drug delivery from the airway lumen. While the clinical development of small-molecule CFTR modulators that are applied systemically, enter airway cells via the basolateral membrane and act on the CFTR protein inside the cells was highly successful, previous attempts to modulate ion channel activity with inhaled drugs that have to engage with their target at the apical surface, such as the P2Y2 receptor agonist denufosol or inhibitors of the epithelial sodium channel ENaC, failed in clinical trials in CF (Graeber & Mall, 2023; Mall, 2020). The results from Guidone and colleagues indicate that the mucus barrier that is characteristically thickened due to impaired mucociliary clearance and increased mucus secretion triggered by the chronic inflammatory milieu in CF airways may have played an important role in the failure of successful translation of these therapeutic approaches from the bench to the clinic. Therefore, future preclinical testing of inhaled drugs should include models with a pathophysiologically more relevant mucus barrier, e.g. by inducing mucus hypersecretion in airway epithelial cultures with proinflammatory stimuli, or by using large-animal models such as the CF pig or the CF ferret, which develop airway mucus obstructions and may thus help to determine to what extent an inhaled drug can overcome a CF-like mucus barrier. Besides inhaled small-molecule drugs, penetration of the mucus barrier is also a major hurdle for the development of inhaled nucleic acid-based therapies (NABT) including mRNA replacement, DNA replacement and antisense oligonucleotides, as well as gene editing approaches, that are currently pursued to address the high unmet need of the remaining ~10% of CF patients who are not eligible for CFTR modulators (Graeber & Mall, 2023). Currently, there are several cargo strategies for NABTs and gene editing in the development pipeline, including lipid-based nanoparticles and viral vectors, which will need to achieve high penetration rates through a thick mucus barrier for effective targeting of airway epithelial cells.

Linked articles: This Perspectives article highlights an article by Guidone et al. To read this paper, visit https://doi.org/10.1113/JPXXXXXX.

The Journal of Physiology

In this context, the study by Guidone et al. also offers a promising solution for how the access of inhaled therapies to the airway epithelium can be improved by removing the mucus barrier with mucolytically active reducing agents (Guidone et al., 2025). DTT will not be suitable for this purpose due to its toxicity and there are currently no effective mucolytics available in the clinical arena. However, a novel thiol-saccharide mucolytic (MUC-031) showed promising effects in sputum from CF patients and in a mouse model with CF-like lung disease, and is currently in early-phase clinical development (Addante et al., 2023). If successful, such novel mucolytics may not only become important as a symptomatic therapy to tackle mucus plugging, but may also help to facilitate the delivery of targeted inhaled therapies to the airway epithelium in CF and other muco-obstructive lung diseases.

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

## Additional information

### Competing interests

M. A. M. reports grants or contracts from, the German Ministry for Education and Research (BMBF), the German Research Foundation (DFG), Boehringer Ingelheim, Enterprise Therapeutics and Vertex Pharmaceuticals with payments made to the institution; personal fees for advisory board participation or consulting from Boehringer Ingelheim, Enterprise Therapeutics, Kither Biotech, Pari, Splisense, and Vertex Pharmaceuticals; honoraria for lectures from Vertex Pharmaceuticals; travel reimbursement for participation in advisory board meetings for Boehringer Ingelheim and Vertex Pharmaceuticals; and a patent describing the $\beta$ENaC-tg mouse as an animal model for chronic obstructive pulmonary disease and cystic fibrosis.

### Author contributions

A.B.: Conception or design of the work; Drafting the work or revising it critically for important intellectual content; Final approval of the version to be published; Agreement to be accountable for all aspects of the work. M.M.: Conception or design of the work; Drafting the work or revising it critically for important intellectual content; Final approval of the version to be published; Agreement to be accountable for all aspects of the work.

### Funding

M.A.M was supported by grants from the German Research Foundation (CRC 1449 – project 431 232 613) and the German Federal Ministry of Education and Research (82DZL009C1 and 01GL2401A).

### Supporting information

Additional supporting information can be found online in the Supporting Information section at the end of the HTML view of the article. Supporting information files available:

**Peer Review History**

