## [Peer Review History · The Journal of Physiology]

Don't forget the mucus barrier in pulmonary drug delivery!

Anita Balázs and Marcus A Mall
DOI: 10.1113/JP288762

Corresponding author(s): Marcus Mall (marcus.mall@charite.de)

The following individual(s) involved in review of this submission have agreed to reveal their identity: Luis J.V. Galletta (Referee #1)

Review Timeline:

Submission Date:	19-Mar-2025
Editorial Decision:	26-Mar-2025
Revision Received:	31-Mar-2025
Accepted:	03-Apr-2025

Senior Editor: Peking Fong

Reviewing Editor: Péter Hegyi

Transaction Report:

Dear Dr Mall,

Re: JP-P-2025-288762 "**Don't forget the mucus barrier in pulmonary drug delivery!**" by Marcus A Mall and Anita Balázs

Thank you for submitting your manuscript to The Journal of Physiology. It has been assessed by a Reviewing Editor and by 1 expert referee and we are pleased to tell you that it is acceptable for publication following satisfactory revision.

The review comments are copied at the end of this email.

Please address all the points raised and incorporate all requested revisions or explain in your Response to Referees why a change has not been made. We hope you will find the comments helpful and that you will be able to return your revised manuscript within 2 weeks. If you require longer than this, please contact journal staff: jp@physoc.org.

REVISION CHECKLIST:

We look forward to receiving your revised submission.

Yours sincerely,

Peying Fong
Senior Editor
The Journal of Physiology

REQUIRED ITEMS

- Please ensure that the Article File you upload is a Word file.

EDITOR COMMENTS

Reviewing Editor:

Comments to the Author:

Very nice work. Well done! Only one comment: the authors have previously made significant contributions to the study of MUC5 in other organs, such as the pancreas, demonstrating their expertise in the field. It may be worth briefly mentioning whether the mechanisms identified in those contexts could also be relevant here, as this would further strengthen the rationale regarding the role of the mucus barrier.

Senior Editor:

Comments to the Author:

Review of your Perspectives piece "Don't forget the mucus barrier in pulmonary drug delivery!" is now complete. As you will read from the attached comments, both the Referee and the Reviewing Editor compliment your treatment of the important topic studied by Guidone et al in their recently accepted manuscript. Both do make suggestions for improvement that you may wish to incorporate in your revised manuscript. I think you should be able to address these points readily, at your discretion.

Thank you for beautifully highlighting Guidone et al's research article. I hope that you too will consider submitting future work to The Journal of Physiology.

REFEREE COMMENTS

Referee #1:

The Perspective by Balazs and Mal is an interesting comment on the study by Guidone and coll. I have only a suggestion:

Page 1, line 2 from bottom: I would add the combination IL-17A/TNF-alpha as the another treatment that increases the mucus barrier. It is actually IL-17A/TNF-alpha that mostly decreases the efficacy of the CFTR inhibitor compared to IL-4.

Page 2, line 10: there is probably an "of" missing before "the mucus layer"

END OF COMMENTS

Response to Referees

Reviewing Editor:

Only one comment: the authors have previously made significant contributions to the study of MUC5 in other organs, such as the pancreas, demonstrating their expertise in the field. It may be worth briefly mentioning whether the mechanisms identified in those contexts could also be relevant here, as this would further strengthen the rationale regarding the role of the mucus barrier.

Response:

We thank the Reviewing Editor for this comment. Due to length limitations, we focused our perspective article on the drug delivery aspect of the luminal mucus barrier, which poses a clinical challenge for inhaled therapies. Because luminal drug delivery in the pancreas is unfortunately not a clinical reality (yet), we did not include references of our previous work that deals with the mucus barrier in chronic pancreatitis.

Referee: #1:

Page 1, line 2 from bottom: I would add the combination IL-17A/TNF-alpha as the another treatment that increases the mucus barrier. It is actually IL-17A/TNF-alpha that mostly decreases the efficacy of the CFTR inhibitor compared to IL-4.

Page 2, line 10: there is probably an "of" missing before "the mucus layer"

Response:

Thank you for your suggestions, we included these changes in the manuscript.

Dear Professor Mall,

Re: JP-P-2025-288762R1 "**Don't forget the mucus barrier in pulmonary drug delivery!**" by Anita Balázs and Marcus A Mall

We are pleased to tell you that your paper has been accepted for publication in The Journal of Physiology.

Yours sincerely,

Peying Fong
Senior Editor
The Journal of Physiology

If you would like to receive our 'Research Roundup', a monthly newsletter highlighting the cutting-edge research published in The Physiological Society's family of journals (The Journal of Physiology, Experimental Physiology, Physiological Reports, The Journal of Nutritional Physiology, and The Journal of Precision Medicine: Health and Disease), please click this link, fill in your name and email address and select 'Research Roundup':

<https://www.physoc.org/journals-and-media/membernews>

- You can help your research get the attention it deserves! Check out Wiley's free Promotion Guide for best-practice recommendations for promoting your work at: www.wileyauthors.com/eeo/guide. You can learn more about Wiley Editing Services which offers professional video, design, and writing services to create shareable video abstracts, infographics, conference posters, lay summaries, and research news stories for your research at: www.wileyauthors.com/eeo/promotion.

The Corresponding Author will receive an email from Wiley with details on how to register or log-in to Wiley Authors Services where you will be able to place an order

EDITOR COMMENTS

Senior Editor:

Comments to the Author:

Thank you for addressing these remaining details, and for contributing your insights to The Journal of Physiology.